# Assessing Climate Adaptation and Flood Security Using a Benchmark System: Some Swedish Water Utilities as Good Learning Examples

Nasik Najar [1,*] and Kenneth M. Persson [2]

1 Department of Construction Engineering and Lighting Science, School of Engineering, Jönköping University, P.O. Box 1026, S-551 11 Jönköping, Sweden
2 Division of Water Resources Engineering, Lund University, P.O. Box 118, S-221 00 Lund, Sweden
* Correspondence: nasik.najar@ju.se; Tel.: +46-736-556-581

**Abstract:** The 2020 Sustainability Index (SI), a benchmark system, shows that 2% of the 184 municipal water and wastewater utilities (WWS) in Sweden have a good performance level (green), i.e., they meet all benchmark requirements for the "climate adaptation and flood safety" ("CA and FS"). In this study, ten Swedish WWS organizations were selected and studied in depth. The goal was to present them as good learning examples to inspire other utilities seeking to improve their results and to clarify and concretize the driving factors, strategies, and important explanations for their success, and the challenges they face. A total of 52 SI annual documents from ten utilities were analyzed. Six of their managers were interviewed in depth. One of the ten utilities studied was green on the parameter "CA and FS". Flooding events in two utilities drove climate adaptation. The formation of an interdisciplinary group in two organizations had a major impact on their success. Two utilities focused on low-lying sites and enclosed spaces. Three believed capacity building increased their chances of success. The biggest challenge was sharing responsibility. That only 2% of municipal water and wastewater utilities are green is not the whole truth. Therefore, there should be other factors in SI that measure performance in "CA and FS". To improve the results, new laws are needed to solve the problem of shared responsibility.

**Keywords:** action plan; benchmark; climate adaptation; flood security; Swedish water utilities; shared responsibility; vulnerability analysis

## 1. Introduction

Cloudbursts will increase by 10–40% in the future, and what is now classified as 100-year rainfall may double in the future if rainfall intensity increases by 25% [1]. An increase in the amount of rain in a short period of time, i.e., caused by cloudbursts, poses great risks of flooding and landslides [2].

In the last 15 years, there have been many severe cloudbursts that have affected many Swedish cities. They occur geographically randomly throughout Sweden and can have a very local distribution [3]. The worst cloudburst occurred in August 2014 in the city of Malmö. The highest rainfall amounts were recorded then in the Centre of Malmö with over 120 mm of rain in 6 h, while the municipality of Staffanstorp, 15 km east of Malmö, was spared from the downpour [4].

According to Swedish design standards, the wastewater system must have the capacity to handle a 10-year rainfall event, which corresponds to ~40 mm of rainfall in 6 h [5]. However, the analysis shows that extreme events cannot be handled only through pipeline construction, as this system is not designed to handle this abundance. It also does not make economic sense to size for extreme volumes, as they occur far too infrequently. However, extreme events must be managed through proper spatial planning and urban

design [5]. According to [6], spatial planning provides tools and processes that can facilitate the implementation of sustainable urban water management (SUWM) [6].

Urban densification and climate change exacerbate the flooding problem. Another factor is the type of wastewater system. In general, areas with a combined system are more affected by flooding than areas with a separate system [7]. The progression of floods causes great damage to private and public property and can pose a direct threat to life. In addition, the cost of flood damage is significant. For example, the cost of the severe cloudburst that hit Malmö in August 2014 was estimated at about 60 million euros [4].

Climate-related risks can be reduced by, among other things, accelerating and increasing the number of cross-sectoral and multifunctional climate actions [8] and adapting cities to potential cloudbursts now and in the future [9], and cloudburst mapping can also help identify where there is a risk of flooding in the community [9].

Municipalities should therefore implement long-term urban water management (L-T UWM) to minimize the risk of urban flooding during large peak runoff events [10]. However, L-T UWM and climate adaptation planning is only lightly regulated in Swedish national Policy and is thus mainly applied by individual water utilities [11,12]. Few municipalities in Sweden are therefore working in L-T UWM [12]. Many cities have explored the possibilities of multifunctional stormwater management that integrates multiple components to manage runoff [13]. However, to create the conditions for multifunctional management, decisions on how to manage stormwater must be made at an early stage, i.e., in the general plan and in the detailed plan [14].

Swedish municipal water utilities (SMWU) play an essential role in climate adaptation, as their planning monopoly gives them a great responsibility for implementing concrete measures [3]. However, the measures should be adapted to the new requirements, according to the new circumstances that climate change has created [3]. The new requirements have made the implementation of measures during heavy rainfall not only the responsibility of SMWU, but also a matter of community planning and division of responsibilities among several administrations in the municipality. However, previous research has shown that the division of responsibilities is a complex issue and a barrier to strategic planning for urban water management [15]. This is in addition to the other institutional and legal obstacles [15]. Furthermore, according to [16], the distribution of responsibilities is a major problem, as it is not clear who is responsible for investigations and their costs, what the requirements are for each party, and what rainfall intensity each party must deal with. The study also highlights the need for national guidelines for responsibility sharing [16].

Moreover, Swedish municipal water and wastewater (WWS) organizations and urban planning administrations lack staff and money. This contributes to the lack of time and capital to enable them to become sufficiently familiar with stormwater management in the planning process [14], and results in current problems being prioritized over long-term solutions. Accordingly, the lack of modern stormwater management solutions is an economic problem, and to achieve long-term stormwater management, economic resources are needed first and foremost for the continuous development of stormwater systems [10]. In addition, there is a lack of a national strategy and action plan for working with climate adaptation in Sweden. Many ministries and agencies are involved, but no one is responsible for climate adaptation [3]. The main responsibility for implementing concrete measures therefore lies with municipalities and individual property owners [17].

The use of benchmark systems can facilitate the management of water utilities. The Swedish WWS sector largely uses the Sustainability Index tool (SI) as a benchmark system. SI was developed by the Swedish Water and Wastewater Association (SWWA) in 2014 [18]. It is a key to steering WWS activities towards sustainability in the future. SI is conducted as an annual survey of WWS utilities and consists of 14 parameters and 82 questions (Figure A1 in Appendix A) [16]. The results for SI for 2020 show, as in previous years, that the parameter "climate adaptation and flood security" ("CA and FS") is one of the two most challenging parameters for the WWS sector. The second challenging parameter is "the status of WWS fixed facilities" [19]. The results for "CA and FS" show that 55%

of the participating municipalities are red on this parameter (i.e., their results need to be improved) and only 2% are green. The main reason for the high red percentage is that the WWS utilities lack a vulnerability analysis and a flood protection strategy for new and existing buildings [19]. In 2021, a survey was conducted to map the Swedish municipalities' systematic work on climate adaptation and to obtain an overview of how the work on climate adaptation was progressing at the local level in Sweden. The results showed a wide dispersion among all municipalities. Of the 180 municipalities that participated in the survey, 8% scored more than 30 out of a total of 33 points. Half of the municipalities did not score 16.5 points, and 26% scored less than ten points in the survey [2]. Accordingly, the improvement activities underway in the municipalities vary, and all municipalities have different conditions for their implementation [19].

In consultation with the Swedish Water and Wastewater Association (SWWA), ten WWS organizations were selected that had demonstrated good capabilities in improving their sustainability outcomes by working with L-T UWM and creating conditions for integrating multiple components. The objectives were to study them in depth and present them as good learning examples to inspire other utilities seeking to improve their outcomes. The research questions that guided the study were as follows: (1) What is the impact of organizational form on utility performance? (2) What factors and strategies have contributed to progress? (3) What are the main explanations for their success? (4) How do they deal with the various challenges, difficulties in sharing responsibilities, and finances when it comes to stormwater management? and (5) Do they have a vulnerability analysis with an action plan?

## 2. Literature Review

*Flood Risk Management: A Shift from Purely Sectoral to Integrated Thinking*

Much of the research has focused on how climate change has created conditions for the development of new knowledge, thinking, strategies and technologies, and how the application of these strategies has in turn created new challenges. These challenges, in addition to the challenges posed by the cloudbursts themselves, are a major barrier to progress in climate [8]. The management of storm water in urban areas was primarily through piped systems. One consequence of this management practice is that many areas are inadequately protected against heavy rainfall. Thus, to achieve better protection against expected climate change and population growth, there should be a shift from a purely sectoral approach to a sustainable approach of integrating urban planning [20]; i.e., reducing the piping of stormwater, creating floodplains, and other open solutions that can absorb excess stormwater during floods without major consequences [21]. Several approaches for decentralized solutions were developed. They have different values according to where they were first developed [22]. Best Management Practices (BMPs) [23,24], Sustainable Urban Drainage Systems (SUDS) [25], and Innovative Stormwater Management (ISM) [26] are some common concepts that were developed.

The term BMPs usually refers to structures that imitate the natural hydrologic processes. The ISM concept includes innovative approaches to mitigating the risk of flooding and reducing pollution impacts [26].

From the 1960s to the 1980s, the traditional flood control approach involved the concept of resistance i.e., reducing the effects of flooding through physical flood protection [27]. The strategy is about keeping water away from land, e.g., by building embankments and raising them continuously [28].

Since the damage can be catastrophic if flood controls fail [28] the traditional flood control measures are accordingly an inadequate method to prevent the growing risks of floods [29]. The newer approach is resilience, i.e., "focusing on risk management instead of on hazard control" [15] takes the possibility of flooding into account and aims thus at minimizing the consequences of flooding. Hence, resilience adapts land-use to reduce flood damage potential; elevating housing structures is an example of its application. Thus, in the context of urban flooding, resilience means robustness, adaptability, and

transformability [30]. Adaptation is often organized around resilience as bouncing back and returning to a previous state after a disturbance [8].

The approach of flood risk management includes thus a shift from purely sectoral to integrated thinking, or in other words, from pure water management to a more resilience-based approach of integrated urban planning to keep vulnerable land uses out of flood-prone areas [20]. This can thus be thought a promising approach to deal with the unpredictability of climate change and future flood risk in cities.

Accordingly, the application of the concept of a sustainable integrated urban water planning (SIUWP) depends not only on access to technical solutions, but also on understanding how to manage them [31].

## 3. Materials and Methods

In consultation with SWWA, ten organizations were selected to be studied through case studies. These organizations had made good priorities and improvements in their pursuit of sustainability in the SI parameter "CA and FS", which is one of the top two challenges for WWS organizations nationwide. A text explaining the purpose and intended study methodology were sent to the selected organizations for their consent if they wanted to participate in the study. After receiving their consent, we also received, upon request, the detailed documents of their SI annual evaluation for the years 2015–2020. Through these documents, we were able to compile the results of the municipalities in the parameter "CA and FS" both at the level of the questions and at the level of the parameters, determine their improvement trend over the years (Section 4.1), and analyze the trends and conditions exhibited by the organizations related to the parameter (Section 5.1). Based on these results, seven organizations that had achieved the most improvements in the parameter were selected for further questioning in the form of in-depth interviews. The interviews, conducted with six out of seven WWS managers, were conducted in October 2021. The final step was to transcribe, summarize, and analyze the interviewees' responses. Figure 1 shows the methodological structure of this study. Figure 1 also shows that the case studies are presented in Section 4. They include, in Section 4.1, a description of 52 documents on SI-detailed evaluation for the ten selected organizations for 2015–2020. In Section 4.2, the in-depth interviews with WWS managers for six of the seven selected organizations are presented. The results and discussion are summarized in Section 5. They include Sections 5.1–5.6. In Section 5.1, an analysis of the results of the 52 documents of the SI evaluation is presented, followed by the result and discussion of the research questions in Sections 5.2–5.6. Finally, the conclusion is presented in Section 6, as visualized in Figure 1.

### 3.1. Background Materials

3.1.1. SI-Tool

The SI is key to guiding WWS operations toward sustainability in the future, and is thus part of the work towards continuous improvement. SWWA developed it in 2014. The SI survey consists of a questionnaire that municipal WWS organizations have responded to since 2015. The survey consists of 82 questions of three main topics grouped under 14 parameters. These groups form the three pillars that interpret the aspects of the Brundtland Commission's definition (BCD) of sustainability. The three pillars, the fourteen parameters, and the code for the 82 questions are presented in Figure A1 in Appendix A [16].

3.1.2. The Parameter "Climate Adaptation and Flood Security" "CA and FS"

"AC and FS" is one of 14 parameters in the survey SI. The parameter is defined at SI by three detailed questions (Ta1–Ta3) (Table 1). The questions cover all measures that need to be taken to protect and adapt both existing and newly planned buildings to withstand change and torrential rains. These measures include: (1) conducting the necessary vulnerability assessments for existing buildings to improve long-term safety; (2) identifying the number of basement floods in a community, as this can provide information on how well municipal drainage systems are functioning for existing buildings; (3) considering proper elevation

when designing new buildings, and having solutions and plans in place for draining stormwater in the ground during extreme weather conditions. Thus, the answers to the questions explain the status and climate adaptation measures that the WWS organization is taking to ensure the future of existing and newly planned buildings. Table 1 shows the questions and the conditions required to score them: green means good, yellow means needs improvement, and red means must be improved. The parameter itself is rated as green if the answers to all three questions are green [18].

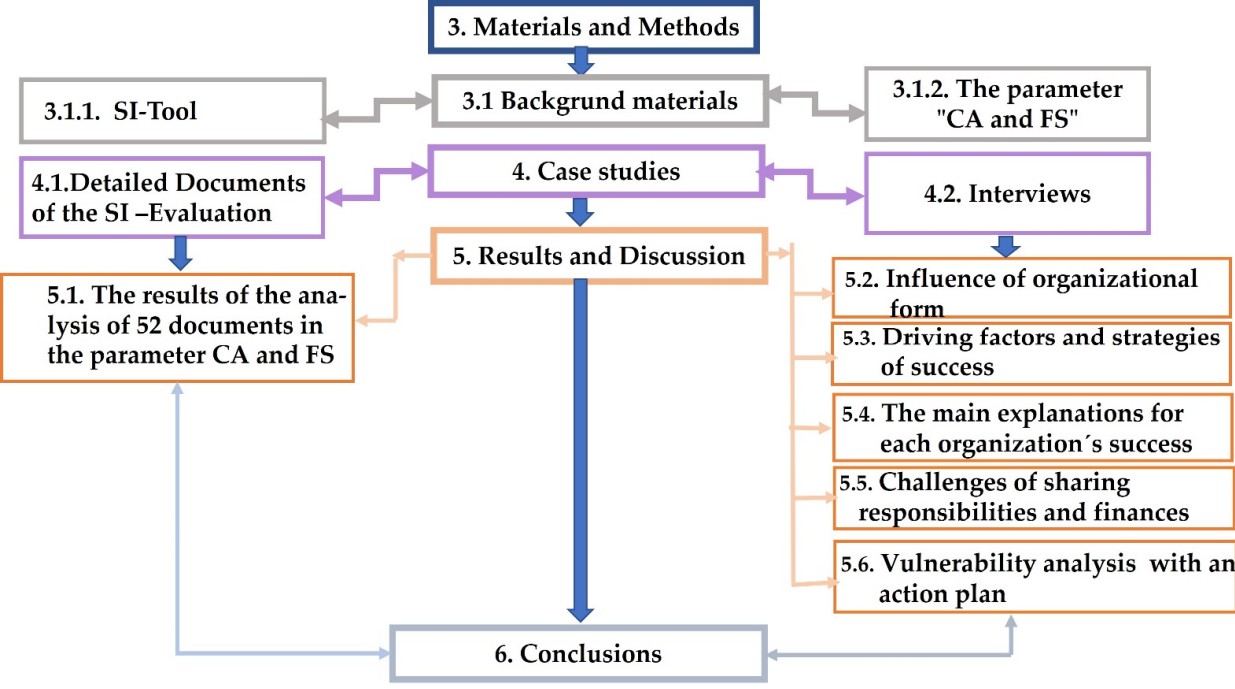

**Figure 1.** Methodological structure of the study.

**Table 1.** The questions for the parameter "CA and FS" and the conditions required for their evaluation [18].

| The Code | Parameter: Climate Adaptation and Flood Security | Conditions for. "CA and FS" | | |
|---|---|---|---|---|
| | | Green | Yellow | Red |
| Ta1 | Is there an investigation with an action plan examining society's vulnerability due to more extreme rainfall and rising levels in seas, water courses, and lakes? | Yes, there is an action plan, and work is being conducted according to this. | Yes, but there is no action plan. | No |
| Ta2 | Is there a clear strategy for the new construction/reconstruction of flood-safe houses and correct height adjustment so that there is no damage to houses when the stormwater systems are overloaded? | Yes, and no floods Can occur in new areas due to rain or water levels. | Yes, but they should have been sharper. | No |
| Ta3 | Basement floods within business areas as a 5-year average (the number per the coupling pipes in 1000 houses) | <1 | 1–2 | >2 |

The SI national results for the parameter "CA and FS" in 2020 show that 55% of the total 184 participating municipalities are red on this parameter and only 2% are green, i.e., 4 organizations out of 184. The fact that a large number of the municipalities are red on this parameter is mainly due to the lack of a vulnerability analysis and flood protection strategy for new and renovated buildings [19].

## 4. Case Studies

Sweden is divided into 21 counties, and each county is subdivided into several municipalities, which total. Each county has a County Administrative Board, which is a state authority that ensures that the decisions of the government and parliament are put into practice. The County Administrative Board ensures that municipalities in the county comply with certain environmental, building, housing, and safety laws and regulations, among others. The municipalities are responsible for a large number of societal services [32].

Ten municipal WWS organizations were selected for this study. Starting with the smallest municipality, they were Arvika, Ljungby, Ronneby, Värnamo, Ängelholm, Luleå, Växjö, Jönköping, Västerås, and Linköping (Figure 2). Figure 2 shows a map of Sweden and the ten studied municipalities and associated counties.

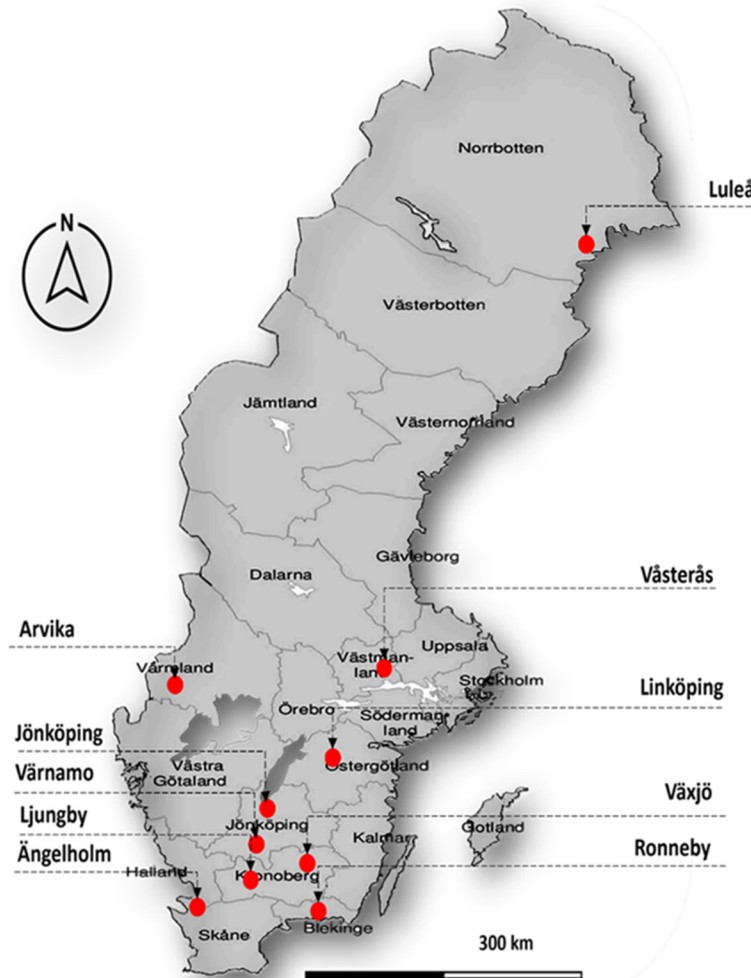

**Figure 2.** Map of Sweden with the names of the ten selected municipalities.

Municipalities in Sweden can be divided into three groups of different size, depending on the number of inhabitants. These are: (A) major cities for municipalities with populations of 200,000 or more; (B) larger cities for municipalities with at least 50,000 inhabitants; (C) smaller cities for municipalities with at least 15,000 inhabitants [33].

Five of the ten selected organizations were in the municipalities of group (C), smaller cities, namely: Arvika (25,865 inhabitants), Ljungby (28,521), Ronneby (29,346), Värnamo (34,030), and Ängelholm (43,030). The remaining organizations belonged to group (B), larger cities. These were Luleå (78,487), Växjö (94,884), Jönköping (142,630), Västerås (155,858), and Linköping (164,684).

Arvika is a municipality in the county of Värmland. It is located on the lake Kylviken, which flows into Lake Glafsfjorden. Glafsforden is a part of the main catchment area of the River Göta.

Ljungby municipality is in Kronoberg county. Lake Bolmen is located in Ljungby and the River Lagan flows through Lake Vidöstern and further through Ljungby.

Ronneby is a municipality in the county of Blekinge. The municipality is located by the Ronneby River in the central part of Blekinge County, with the Baltic Sea and parts of the Blekinge Archipelago to the south.

Värnamo is a municipality in Jönköping County. It is located by the River Lagan, not far from its tributary in the lake Vidöstern.

Ängelholm is a municipality in Skåne County. The River Rönne å, which flows into the Skälderviken, meanders through the city area.

Luleå is a municipality in Norrbotten County. It is located on the Gulf of Bothnia. The Luleå River flows through the municipality from the northwest and empties into the sea in the southwest.

Växjö is a municipality in Kronoberg County. The Växjö Lakes are a contiguous lake system: Barn Lake, Trummen Lake, Växjö Lake, Södra Lake, and Norra Bergunda Lake.

Jönköping is a municipality in Jönköping County, located on Lake Vättern. The old urban area of Jönköping consists of a sandy area with two smaller lakes (Munksjön Lake and Rocksjön Lake) on the southern shore of Lake Vättern, surrounded by hilly slopes.

Västerås is a municipality in the southern part of Västmanland County on the northern shore of Lake Mälaren.

Linköping is a municipality in Östergötland County. The Stångån River flows directly through the municipality and empties into Lake Roxen. Lake Roxen is 10,000 hectares in size, but only seven meters deep. The Göta Canal also runs through the municipality. Figure A2 in Appendix B Shows pictures of the map in all ten municipalities.

Six of the ten organizations studied, namely Värnamo, Ängelholm, Luleå, Växjö, and Jönköping, have a traditional WWS organizational form. The form in Arvika and Ronneby is that of a municipal enterprise, and the organizations in Västerås and Linköping are multi-utility companies. For more information and a description of the different types of organizations, see Section 5.2. "The influence of organizational form on performance."

*4.1. The Detailed Documents of the SI Evaluation for the Ten Studied WWS Organizations*

The detailed documents show the organizations' annual answers and their assessment with a color index for all the 82 questions and the 14 parameters. A description of the documents' contents and structure is presented in Appendix C. An example of one of the 14 parameters in a typical detailed document for SI evaluation is presented in Figure A1 [34].

In response to our request, the ten selected organizations provided us with 52 detailed documents of their SI evaluation for all the years in which they participated in the SI survey. Seven out of ten organizations participated in the 2015–2020 survey. Ronneby municipality, on the other hand, participated in 2016–2020, Ängelholm participated in 2017–2020, and Ljungby participated only in 2020.

The statistical weighting method was used to evaluate the results, i.e., assigning a performance score for the parameter "CA and FS". Since in SI, the answer to each question is given a traffic light color, green (good), yellow (needs improvement), or red (must be improved), a performance score from 0 to 2 was used to calculate the weighting limits of the parameter. By setting a weighting limit of 2 for answers with green color, 1 for yellow color, and 0 for red color, we calculated the weighting limits for the parameter as the average of the weighting limits of all questions belonging to that parameter (Figure 3). Figure 3 thus illustrates the color index valuation of all questions for the parameter "CA and FS", for the year 2020 for all participating municipalities. Figure 3 also shows how the weighting limit is calculated. For example, in row two above, it shows the weighting limit for all ten organizations for the year 2020; it was calculated in the same way for the other years. The

results for the "CA and FS" parameter for all ten WWS organizations surveyed in the years in the SI survey are shown in Figure 4.

| Parameter: Climate Adaptation and Flood Safty "CA and FS" | Arvika 2020 | Ljungby 2020 | Ronneby 2020 | Värnamo 2020 | Ängelholm 2020 | Luleå 2020 | Växjö 2020 | Jönköping 2020 | Västerås 2020 | Linköping 2020 |
|---|---|---|---|---|---|---|---|---|---|---|
| Calculating the weighting limits. Green answer = 2, yellow = 1 and red = 0 | (2*2)+ (1*1)=5/3 = 1.7 | (1x2)+ (2x1)=4/3 = 1.3 | (1*2)+ (2*1) = 4/3 = 1.3 | (2x2)+ (1x1)=5/3= 1.7 | (2x2)+ (1x1) =5/3 =1.7 | (2x2)+ (1x1)=5/3= 1.7 | (1x2)+ (2X1)=4/3 =1.3 | (3x2) = 6/3= 2 | (2x2)+ (1x1)= 5/3 = 1.7 | (2x2)+ (1x2)= 5/3 = 1.7 |
| Ta1:Is there an investigation with an action plan on society's vulnerability due to more extreme rainfall and rising levels in seas, watercourses and lakes? | Yes, but no action plan | Yes, but no action plan | Yes, but no action plan | Yes, but no action plan | Yes, but no action plan | Yes, but no action plan | Yes, but no action plan | | | Yes, but no action plan |
| Ta2:Is there a clear strategy for new construction, reconstruction, for flood-safe and correct height adjustment that there is no damage to houses when the stormwater systems are overloaded? | Yes and no floods can occur in new areas due to rain or water levels. | the strategy should have been sharper. | the strategy should have been sharper. | | | | the strategy should have been sharper. | | the strategy should have been sharper. | |
| Ta3: Basement floods within business areas as a 5-year average (The number per 1000 house coupling pipe) | 0.41 (4 in 5 years) | 0.51 | 0.57 | 0.976 | 0.37 | 0.779 | 0.07 | <1 | 0.32 | 0.57 |

**Figure 3.** Evaluation of all the questions for the parameter "CA and FS" as well as the calculation of the weighting limits of the parameter for the ten studied organizations in the year 2020.

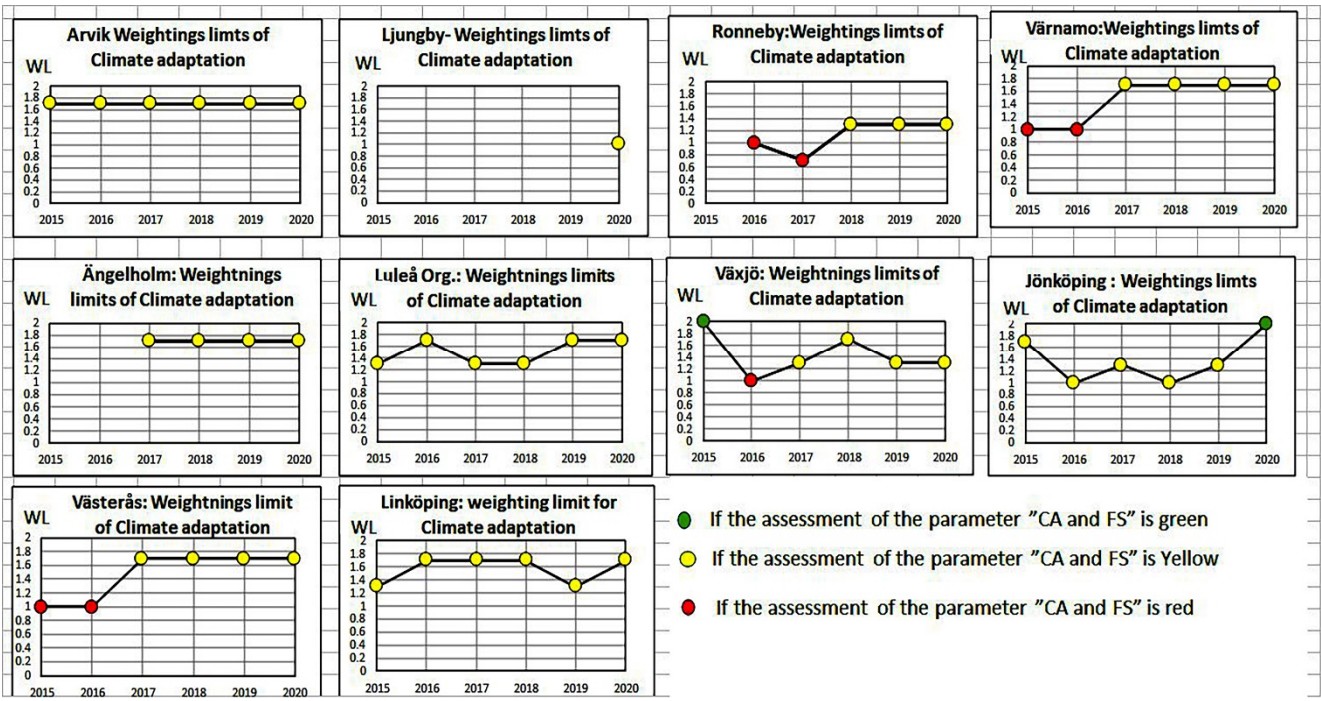

**Figure 4.** Compilation of the results for the parameter "CA and FS" for all ten studied W and WS organizations during their participation in the SI survey.

Figure 4 also shows the results as development trends for the parameter for all ten participating municipalities and the years during which they participated in the SI survey. The analysis of the results shown in Figures 3 and 4 is described in more detail in Section 5.1.

### 4.2. Interviews

Based on the obtained results of the 52 studied documents (Figures 3 and 4), it was decided which organizations were particularly good learning examples. Thus, the organizations that had made the most improvements on the "CA and FS" parameter were selected and surveyed further. Seven out of ten organizations with weighting limits of at least 1.7 on the parameters "CA and FS" were selected for in-depth interviews. These organizations were Arvika, Värnamo, Ängelholm, Luleå, Jönköping, Västerås, and Linköping. Linköping,

though, did not have the opportunity to participate in the survey. Semi-structured interviews were conducted with the remaining six leaders of the WWS organizations in October 2021. Table A2 in Appendix D provides the names of the six communities in which the selected WWS organizations operate, the number of residents in the communities, the first initials of the names of the managers of the WWS organizations, the number of years they had spent as managers, their number of years in the WWS sector, and the type of organization.

Twenty interview questions based on the objectives and research questions of the study were compiled and sent digitally in a file to the managers of WWS organizations, along with a schedule for the interviews. Results from the analysis of the documents of the SI detailed evaluation at the parameter level for "CA and FS" for all years (Figures 3 and 4) were also included in the file. Many of the interview questions began with brief background information. The interviews, which lasted approximately one hour and thirty minutes per organization, were recorded using the Zoom program. The recordings of the interviews were transcribed and controlled.

## 5. Results and Discussion

This section shows and discusses the results that clarify and concretize how "CA and FS" were handled by the good examples studied. The research questions in Sections 5.2–5.6 serve as headings for the presentation of the study's findings and their discussion. Section 5.1 summarizes and discusses the findings of the 52 documents analyzed.

### 5.1. The Results and Analysis of the 52 SI-Detailed Documents in the Parameter "CA and FS"

The diagrams in Figure 4 show how the parameters "CA and FS" have changed over the years in the ten organizations studied. The diagram also shows that the trend in the municipality of Jönköping turned green in 2020; thus, its weighting limit was 2. The weighting limit in the six other municipalities, i.e., Arvika, Värnamo, Ängelholm, Luleå, Västerås, and Linköping, was 1.7, which means that only one of the three questions (Ta1–Ta3) for 2020 was yellow. The reason was that they lacked an action plan (Figure 3).

In the remaining municipalities studied, the weighting limit was 1.4 and below, which means that they were yellow in two out of three questions or in all three questions.

However, Figures 3 and 4 show that one of the study's ten participating organizations was green (i.e., 10%), nine (i.e., 90%) were yellow, and none were red (i.e., 0%) in the "CA and FS" parameter, compared to the national results, in which 2% were green, 43% were yellow, and as many as 55% were red [19].

### 5.2. The Influence of Organizational Form on Performance

Four of the six organizations interviewed, namely Värnamo, Ängelholm, Luleå, and Jönköping, have a traditional WWS organizational form, in which the politicians in a municipal board are ultimately responsible for the activity carried out by the organization [35]. The traditional form is a one of several administration parts under the responsibility of the technical board within the municipality. The technical board thus usually has several issues to deal with that belong to all administrations [36]. The traditional form is the most common form of WWS organization in Sweden [37]. WWS organization in Arvika, on the other hand, is a part of a municipal company, "Teknik I Väst AB", jointly owned with the neighboring municipality of Eda. The organization in Västerås is a multi-utility company, "Mälarenergi AB".

Multi-utility is a term for a company that carries out several activities in the field of technical infrastructure. The company is divided into several business units, so water and wastewater form a separate business unit [38]. When a municipality transfers the operation of WWS organization to a joint stock company, the operational and financial responsibility is transferred to the company [38]. Thus, the role of the municipal council is no longer to approve the budget and set the overall goals and guidelines for an activity. The CEO

is responsible for the operational activities, and the strategic responsibility lies with the board [35].

Active ownership represents an internal relationship, which means that it is the municipality itself that can create the conditions for conducting long-term sustainable plans [37]. It is important for long-term, sustainable operations, regardless of organizational form [37].

In this study, Luelå and Ängelholm, which have a traditional organizational form, were found to believe that their organizational form was of great importance when it came to their positive results. The manager in Ängelholm believed that their decision-making paths were much shorter, that they could plan for the long-term significantly more, and that staff turnover was much lower than in other companies. He also emphasized that they had formed a climate adaptation group, which created excellent opportunities for contacts and collaboration and was the reason for the success at Ängelholm. However, the manager believed that they would not have formed the group if WWS was not traditional form. This is consistent with the case study of Skellefteå [37]. Skellefteå, which also has a traditional organization, showed obvious abilities to develop long-term plans, strategies, and well-functioning processes. On the other hand, Arvika and Västerås emphasized the advantages of the corporate form of the WWS organization. They thought that their organizations Teknik I Väst AB and Mälarenergi AB, were of great importance for their positive results.

The manager of WWS and road departments in the company -Teknik I Väst AB- also believed that "Teknik I Väst AB", which included all technical activities in the municipality, i.e., WWS, electricity networks, etc., was of great importance for its success. He believed that part of the political board, which also sat on the municipal council, had focused on the right thing, as they could only consider the technology and could not mix other issues, such as school, care, etc., as in the traditional form. The company had a very good impact and opportunity to obtain more competent resources, which is in line with [35]: "The main reasons behind choosing a new organizational form are to enhance the competence and ensure more resources and efficient use of them". The manager also said that before they started the company, they had difficulty in obtaining investment funds to run WWS. This is consistent with [39]: "Incorporation creates clearer financial accountability by separating the income statement and balance sheet of operations from other municipal operations".

The manager and the investigator at Västerås also believed that their multi-utility company had contributed to faster decision making, which was also emphasized by [40]: "that decision making paths are becoming shorter and the decisions of a strategic nature are being made closer to the company". The manager also believed that the WWS organization had a very strong position in the municipality. This was because, as also claimed in [35], the CEO and the part of the board that represented the company in the municipal council had a clear mandate and responsibility for the company. This also helped to ensure that they had a clearer place in the community and were heard for the company's concerns [35].

*5.3. Driving Factors and Applied Strategies That Have Contributed to the Organization's Success in the Field of CA and FS*

Two significant events in Arvika have driven climate adaptation toward sustainability. The first was a major flood in 2000, which began with the rising of the Lake Glafsfjord to more than 3 m, and flooded parts of the lower part of the city. The second incident was the flooding of basements over several years starting in 2006, which caused an outcry among the population. A similar event was the reason for the development of the climate adaptation process in the municipality of Värnamo, when in 2004 the banks of the river Lagan, which flows through the whole of Värnamo, overflowed, causing significant consequences for the community. "After the flooding, we dealt with problems on all fronts", said the manager in Värnamo. This coincided with the statement of a WWS manager from [16] that "the problem is not taken seriously, and resources are not available, until the community is really affected by flooding".

These events have also shown the vulnerability of the existing stormwater management systems, which are in line with [2]: "Analysing previously occurred extreme weather events can be important for understanding the vulnerabilities and vulnerable areas that exist in the municipality".

Consequently, Arvika has developed the Climate Proof Area project. The project is an applied strategy and is a driving factor for success in the field of "CA and FS". The project conducted sizing and capacity calculations for stormwater pipes throughout the center of Arvika, and provided insights into the capacity and condition of the network. Measures included replacing pipes with larger ones, but not replacing pipes that were 20 years old, even if they were small. Many other measures were taken, some of which were proposals from contracted consultants. Half of the consulting costs for the EU project Climate Proof Area (CPA) were covered by the EU, the manager in Arvika said.

The most likely reason for Ängelholm's success was the formation of an interdisciplinary climate adaptation group within the municipality, as the head of WWS organization explained. The group included representatives from the WWS side, as well as the city planning, street, park, environment, and building permit sides. Climate adaptation actions were overseen through the working group, regardless of whose specific areas they affected. The climate adaptation group also had a steering group that determined the direction. For example, the focus may be on increasing the number of wetlands. The group did a lot to make things work so well in the municipality of Ängelholm because it found natural contact areas between the different units and administrations. The manager in Ängelholm believed that working closely together was a good tool.

A crucial factor for the success in Värnamo was the creation of data models on stormwater pipe networks and the level of the Lagans River, which flows through the city. In these data models, measures were simulated to see what impact a measure, such as a protective dike, could have on the Lagan's runoff, and how the flood would look with a dike installed.

According to the manager in Luleå, one of the factors that helped the WWS organization make a good start was that they adopted the climate guidelines from 2015. As a result, there were now anchored plans for overall sustainability. The development of the SI had also been an important factor contributing to the success in Luleå, according to its manager. Moreover, the development process was based on the current situation. Thus, the management team set its priorities based on the results of the SI, which was very helpful for showing the current situation. Given that, one of the goals of the development of SI was "to use the results of the SI survey to enable systematic tracking of municipality improvement activities" [18].

The WWS in Jönköping was one of 4 out of 184 national organizations that received a green grade in terms of the parameter "CA and FS". Jönköping's success was partly due to its focus on flooding, including cloudburst mapping of the central city area and locating low-lying areas in it. The goal of locating low-lying areas was to determine how runoff would appear in the event of a hundred-year rainfall, according to the manager at Jönköping. This is in line with [41]:"cloudburst mapping can successfully be used in risk assessments". Cloudburst mapping is a comprehensive management process involving the rescue service, and the WWS department is heavily involved in the process with its expertise in runoff modelling, etc., added the manager at Jönköping.

In addition, the manager went on to say that "in connection with the work on the cloudburst mapping, a northern storm was counted in Lake Vättern when it was pouring in with heavy rain". However, Jönköping is located in the southern part of Lake Vättern. They expected Lake Vättern to tip, because the land uplift in the northern part (about 2.7 mm/year) is greater than in Jönköping at the southern tip of Lake Vättern (about 1.3 mm/year). This means that the level of Lake Vättern in Jönköping increases by about 1.4 mm per year.

"Several cloudburst maps were made in different areas and soon it will be time to make new ones, because circumstances have changed and there has been a lot of rebuilding", said the manager.

Another driving factor in Jönköping was developing criteria for replacing the pipes. Jönköping had also had a renewal strategy since 2012, and they continued to work on it. In addition, measures had been taken for the most vulnerable existing buildings in low-lying and confined areas. This coincides with the statement of [42] that "Lowland areas in previously developed areas are most vulnerable to flooding. The focus should be on measures in these critical areas to reduce the impact and risk of damage".

The WWS in Jönköping had also succeeded in creating a long-term plan and gaining full political support by informing politicians about what exactly the organization needed to do and why. "Political support in the form of investment funds and support in implementing measures has thus been the key to our success," said the manager in Jönköping.

The manager in Västerås explained that their multi-utility Mälarenergi AB Group had an ownership policy that clearly stated that it must operate sustainably. They had thus set sustainability targets in the strategy and development plans that they worked with and followed.

As can be seen from the above, the driving factors and strategies employed by the organizations in this study were characterized by more holistic approaches to risk management that focused on the consequences of flood hazards. This included, as [20] noted, "a shift from water-only management to a more comprehensive approach that incorporated urban planning to keep vulnerable land uses out of flood-prone areas." Moreover, most of the actions taken in this study confirmed that key events put urban water issues on the political agenda and initiated further systematic changes, as in the case of Skellefteå [12]. They also showed that political support in the form of investment funds and implementation support had been key to success. This is also consistent with [43]: "key events can help planners change local decision-makers'—politicians'—understanding of this issue in relation to other urban issues."

According to [21], "reducing the mixing of rainwater and wastewater is one of the protective measures against future climate change." It is also one of the measures that belong to flood risk management.

WWS organizations are attempting to convert their combined systems to dual systems, where wastewater and stormwater are diverted into their respective pipes to reduce flooding. Today, 13% of the sewers in Sweden are still combined sewers [44].

Arvika and Ängelholm have converted their entire combined sewer system and have nothing left; 3% of sewers in Västerås are combined systems; 3–4% of sewers in Värnamo are combined systems; and 10% of sewers in Luleå are combined systems. Jönköping, on the other hand, has 18%, which is above the Swedish average.

*5.4. The Main Explanation for the Organization's Success*

When the managers of the six organizations were asked about the most important explanation for their success, they gave different answers.

The manager in Arvika believed that one of the main reasons for their success was that they had formed a technical staff after moving to corporate form in 2011. The technical staff group housed metrologists, technicians, and engineers. Thus, all the expertise was on a separate staff with its own manager, and the group was not as tied into the operation organizationally. "The group provided support as an internal consulting firm and that was great and a good way to attract competent people to us" said the manager of the WWS and Road departments in the company "Teknik I Väst". He added that "politicians support us and recognize that we need to invest and that would not affect the collective tax." Moreover, they had applied different solutions depending on the conditions on the ground: they had rehabilitated the two low-lying areas in Arvika, they provided specific properties with check valves and built storm-water magazines, and they carried out roof inventories and used a lot of resources and money to achieve better security and at least meet the legal requirement for a rainfall intensity of ten years.

The success in Ängelholm, as in Arvika, was due to the formation of a group, i.e., the climate adaptation group, according to the manager in Ängelholm. "Members of the

group are very interested in their tasks, which are to identify problem areas, make decisions and take action. For example, if there is always flooding in a particular creek, they might propose that a wetland less prone to flooding could fit there instead. The proposal is discussed in the group, funds are requested for the proposal and eventually there will be a wetland there because everyone is working toward the same goal", the manager in Ängelholm said.

Building competence in the WWS organization of Luleå was an explanation for their success, said the manager, explaining, "We have a long history of cooperation with Luleå University of Technology (LTU) and have funded a PhD student in stormwater. We are also part of a SWWA R&D cluster (Dag and Nät) working on stormwater and pipe network topics. We also have staff who work half with us and half with LTU. They are funded by the cluster and work on renewal planning, among other things." This is consistent with [3], which states that "sufficient human resources and competent and knowledgeable personnel are required to operate WWS organizations in an environmentally sound manner". The WWS organization also works with water levels in lakes. "The SI survey has also been very helpful. It has allowed the organization to prioritize and focus on what others are doing," said the manager at Luleå.

Värnamo attributed their success to the understanding and curiosity of politicians and to the fact that the WWS organizations had been good at applying for external funding; for example, they received funding from the Local Nature Conservation Initiative (LONA grant) and the Local Water Conservation Project (LOVA grant) to build wetlands and dams. In addition, Värnamo employees had good skills and a strong interest in pursuing these issues together, according to the manager in Värnamo.

Jönköping also attributed its success to political support for its long-term planning: "This means," explained the manager, "that we have received both investment funds and support in implementing measures. We have received political support by informing politicians about what we intend to do and why. For example, reducing basement flooding. We have received support even though we are causing major problems for residents and business owners by digging up the downtown area. Another reason for our success was a new climate adaptation policy for the community that replaced the old one. It was a clear strategy with action plan," added the manager in Jönköping.

According to the manager in Västerås, the main explanation for their success was also that they had been working with long-term plans for many years, that they dared to try things, and that they cooperated with the city of Västerås where there were other administrations that had a shared responsibility with WWS. The employees also showed a very high level of commitment. She added that they had over 150,000 customers that contributed to their good financial situation and allowed them to hire resources. "I have almost a hundred people available with different skills. This can be compared with the neighboring municipality of Surahammar, where they do the same work, but are only six people, so it is a big difference," said the manager in Västerås.

In summary, Arvika and Ängelholm attributed their success to the formation of groups that worked across organizational boundaries and encompassed many disciplines. This is generally consistent with [20]: "the vulnerability of the hinterland can only be reduced if different disciplines such as water management, spatial planning, and disaster management work intensively together". Värnamo and Jönköping, on the other hand, attributed their success to the support of politicians, and Luleå believed that skill development was the explanation for their success.

*5.5. Various Challenges and the Difficulty of Sharing Responsibilities and Finances When It Comes to Stormwater*

Existing storm sewer systems were developed using different principles than those used today, and were designed for lower rainfall intensities. Flood control in these systems cannot be managed without close cooperation between the relevant municipal administrations [3], i.e., those responsible for building permits, municipal planning, parks, roads,

environment, and WWS. This is consistent with the statement in [28] that "flood risk management is becoming a societal task that requires cross-disciplinary cooperation," i.e., a concept of shared responsibility.

Most of the WWS managers surveyed in 2020 [16] considered responsibility sharing to be a complex issue, with many conflicts of interest that required large resources to address strategically. Managers indicated that there was an unclear division of responsibilities for stormwater in the built area, and that there were questions about who should be responsible for both investigation and cost, and what the requirements were. In addition, no one felt that the WWS organization was not the only one responsible for this issue [16].

It was, therefore, interesting to examine this issue again with the good learning examples in this study to show if and how they could deal with the difficulty of dividing tasks.

It turned out that Arvika and Ängelholm felt that they could handle task sharing well. Both organizations stated that it was partly due to their organizational form that they were able to handle the problem. However, despite having two different organizational forms. Ängelholm's manager explained, "I also think it has to do with the fact that the people who are discussing this are usually also part of the climate adaptation group, which we could not have created if we did not have the traditional form of organization".

The manager in Arvika explained that two events that affected Arvika, the cloudburst event in 2000 and the floods in 2006, made people in Arvika aware of the division of responsibilities and how to work together, and also who should pay for what.

Värnamo said they also had no problem sharing responsibility when it came to new investments. "There is money in an investment budget that is covered by the tax collective" the Värnamo manager said. He also explained that the cost of the measures implemented by the WWS department to mitigate the impact of torrential rains on existing areas would also be borne by the community tax. However, "the WWS department will cover the costs if the stormwater network is undersized and we then build, for example, retention basins or dams".

The managers of the other three WWS organizations surveyed, namely Luleå, Jönköping and Västerås, believed that the division of responsibilities was not simple. "We are definitely not yet at the stage where we have a simple formalization and an effective relationship," said the manager in Luleå. "However, there is some clarification in the stormwater plan about how this should be done, but it is especially difficult when it comes to cloudburst management. On the other hand, we have a clear cost-sharing principle when it comes to joint projects between roads, parks and WWS", the Luleå manager continued.

The manager at Jönköping believed that the division of responsibilities was the most difficult challenge they faced. He believed that other departments of the municipality should require the municipality to use tax revenues for the actions for which they are responsible for. Thus, WWS fees would be reserved to fund the work that was the responsibility of the WWS organization. On the other hand, he mentioned a practical event from 2013 when a hundred-year rain fell in Jönköping. As a result, tax money was allocated, and the road and park were responsible for building slightly larger ditches and bumps in the terrain to drain the water away from the hospital area. The measures cost about 80 million SEK, which was paid by the municipality with tax funds. The WWS organization did not have to pay, because the stormwater pipes were sized and well-maintained according to the standard for more than 20 years of rain.

This again confirms that "key events put urban water issues on the political agenda and initiate further systematic changes" [12]. Despite this incident, it was not easy to overcome other difficulties in the division of responsibilities and funding, said the manager in Jönköping. "There are overlaps in the legislation, in the Environmental Code and the Water Services Act, and there are gaps in the legislation regarding stormwater management", he continued, "and it's very unclear what rights we have to make claims and so on. It's also too uncertain and we do not know what requirements we can put on a property owner, either in terms of the quantity or the quality of the stormwater. There can also be double regulation, the WWS principal may say yes to a stormwater discharge and the

environmental agency says no. We should not have double legislation." Furthermore, the manager at Jönköping said, "We now have a stormwater group at management level that is trying to agree and find a common line, but I think we need clearer legislation to make this work."

However, there were a number of "Swedish Public Inquiries" (SOU) that had proposed changes to the legislation and indicated that clearer stormwater legislation is needed [21]. The manager at Jönköping also believed that the government had still not found a solution, and that this had been the case for several years.

Västerås had also failed to come to grips with the difficulties of sharing responsibilities and funding, according to the manager. The investigator in Västerås, similarly to the manager in Jönköping, also emphasized the importance of clarifying the division of responsibilities in legislation to create clear incentives for the work, so that it did not just depend on the commitment of the individual. The manager in Västerås believed that the biggest challenges they encountered on their way to make improvements were difficulties in collaboration, and that from time-to-time, certain actions needed to be put on hold because something else took higher priority. The investigator in Västerås also considered finances a challenge and emphasized that the process of pushing through various investments was still somewhat slow. "At least as a municipal company, we have it a bit easier when it comes to making decisions about investments", said the manager in Västerås.

In summary, three of the six organizations studied believed they had no problems with task sharing. These were Arvika, Ängelholm, and Värnamo. Arvika and Värnamo had in common the fact that they were both affected by torrential rains and floods. These events had thus set climate adaptation measures in motion and put them on the political agenda. According to the WWS manager in Arvika, they were also the reason that those responsible in the different departments had become aware of the division of tasks and that they worked together and knew who paid for what. Furthermore, in Jönköping during the cloudburst and flood event in 2013, all the departments involved took their share of responsibility and took the necessary measures and paid the costs.

In Värnamo, the measures to contain cloudburst damage were financed with taxpayers' money. However, this required the understanding and support of politicians. This is in line what the manager in Värnamo said: " political support in the form of investment funds and support in implementing measures has been the key to our success", and consistent with the statement in [12] "previous studies have shown that floods or extreme rainfall often influence the political significance of urban water issues".

In this study, we found that the problem of shared responsibility is still largely unsolved. However, apart from major events that shake up both politicians and all those responsible, the formation of groups consisting of many disciplines and working in an interdisciplinary way within the municipality has helped Arvika and Ängelholm to solve the problem. The WWS managers in these cases demonstrated an understanding of how to manage the problem, consistent with the statement of [31] that a SIUWP depends not only on access to technical solutions, but also on understanding how to manage them. However, managers in Jönköping and Västerås believed that clearer laws and regulations were needed to facilitate the cooperation and work of the group formed. This is in line with [37]: "unclear national guidelines can create weak incentives for the development of long-term plans, strategies and well-functioning processes."

*5.6. Vulnerability Analysis with an Action Plan due to Extreme Rainfall*

Climate change will have impacts that affect all parts of the water supply and sewer system. These may include changes in groundwater levels and sea levels, as well as rainfall affecting stormwater management and droughts. In Sweden, the total investment needs for climate adaptation to be implemented over the next 20 years are estimated to be SEK 250 million/year for measures in treatment plants and SEK 900 million/year for pipeline networks [44].

Stormwater management should be sustainable [31] and incorporate the resilience approach of integrated urban planning to keep vulnerable land uses away from flood-prone areas [20]. Thus, the measures must include both open solutions and reconstruction of the sewer system. In this context, an important element of resilience—adaptability—should be used [30]. For example, land use is adapted to minimize the potential for damage, such as by increasing the height of residential buildings or by taking measures to defend the hinterland [28]. In addition, modeling calculations are a necessary tool to determine the most cost-effective measure that meets the selected level of ambition [45].

SWWA has called for the creation of an action plan in addition to a vulnerability analysis for existing and newly planned buildings (Figure 3 and Table 1, Ta1). Without having an action plan, the organizations cannot become green in the parameter "CA and FS" in SI [19]. Nationwide, only 4 out of 184 municipalities have an action plan and are thus green in this parameter [19].

The results for eight out of ten municipalities in this study show (Figure 3-Ta1) that they had produced an investigation (vulnerability analysis) but no action plan.

Some reasons why it is difficult to develop an action plan were cited by several WWS managers in a 2020 interview as part of a study [16]. Among the reasons were (a) the unclear division of responsibilities, which leads to uncertainty about who is responsible for developing the action plan; (b) the fact that the issue should be decided in the community organization and not in the WWS organization; (c) it is difficult to find all the actions to create an action plan; and (d) it is also not clear who will fund the actions once you have an action plan [16].

The six good examples we interviewed in this study were also asked to share their views on this topic. Thus, we found that all six organizations had vulnerability analysis. However, three of them, Arvika, Ängelholm, and Luleå, believed that they did not need to create an action plan because it was difficult to create a detailed action plan as it involved a lot of work and also needed to be adjusted frequently. Instead, they had used the time to implement several improvement actions, which are described in Sections 5.3 and 5.4. The manager in Arvika said, "The action plan will need to be revised frequently because something is happening all the time: a pipeline is moved, a new magazine is built. So instead of having a decided action plan, we have done a lot of other things that are perhaps more necessary than what we had noted in the action plan."

The manager in Ängelholm had a similar view to Arvika: "We have a good grip on our network and know where we have weaknesses and strengths. We have planned for five years as well as ten and fifteen years, based on the age of the pipelines, choice of materials, dimensions and operational aspects. We changed our combined system to a dual system throughout the community, so I do not think we have many flooding problems. All of this has meant that we have not really needed an action plan."

Ängelholm and Luleå are also coastal municipalities, so they had good possibilities for discharging water into the sea. The manager in Ängelholm said, "Because of our closeness to the sea, a lot of the water flows into the sea by itself. We also have a large river running through the city, the River Rönne å, which is a huge stormwater pipe."

Luleå was also not worried about the consequences of not having an action plan, because Luleå is also very close to the sea and has good recipients. Thus, these three municipalities gave different reasons for the lack of an action plan than those given by the organizations the study by [16].

The other three organizations, Västerås, Värnamo, and Jönköping are located inland, and have different geographic conditions than the coastal communities and fewer opportunities to drain large amounts of water during heavy and torrential rains. Therefore, it is even more urgent for them to have an action plan.

Västerås thus had an action plan, but it was 10 years old and applied to the entire municipality. It also had shortcomings in terms of specific action plans for WWS activities, such as climate adaptation of drinking water supply or wastewater treatment, etc. "We probably need to make it more specific to WWS," said the manager in Västerås. The

investigator in Västerås added: "The work on the action plan is led by the City of Västerås. Thus, the various administrations in the city are contributing to the work, and WWS is also involved".

Värnamo, on the other hand, was in the process of developing an action plan but still lacked much basic material "Once we have studied how the measures will affect the city, we will create an action plan. The plan will include several measures, including stormwater roads, dikes, detention basins, etc. There is already money earmarked for disaster prevention in an investment budget on the tax side,"said the manager in Värnamo. According to him, "It can be easy to build the wrong things in the wrong places, for example, if you do not have computer models to support your work. In other respects, you need to have a good dialog with politicians".

Jönköping, on the other hand, was the only municipality in this study that had an action plan, and one of four municipalities in Sweden that had an action plan. However, Jönköping had very special geographical conditions because of Lake Vättern, which is the second largest lake in Sweden and provides the municipality with drinking water. Jönköping had therefore conducted a vulnerability analysis with an action plan for the city's exposure to extreme rainfall and the rise of Lake Vättern. The result of this analysis had shown that the level of Lake Vättern can reach a peak of + 90.3 if several unlikely factors occur at the same time, such as a hundred-year intensity rainfall. The level of Lake Vättern was currently at a high of 88.95 [46]. The hazard analysis had shown that at a level of +90.3, two large areas in the municipality, Liljeholmen and Öster, were at risk of flooding. The analysis also showed that a large number of basement floods would occur in these two areas, even with a ten-year rainfall intensity.

The level + 90.3 was thus adopted by the City Council in the Jönköping Climate Adaptation Plan. In the action plan, it was decided to use the +90.3 level as a reference for setting heights for new buildings and low-lying vulnerable areas. The pumping station in the Liljeholmen area had been equipped for this height, and this was also proposed for the Öster area. In addition, barriers would be required in some locations when the +90.3 level was reached.

The supply line of the waterworks was also lowered by a few meters. The drawdown was adjusted to match SMHI climate models for the next 100 years.

However, the difficult part of the action plan was funding the measures against torrential rains. The first steps toward improvement could be taken at the WWS organization level, but the rest must be weighed against all other investment needs on the tax side, according to the Jönköping manager.

## 6. Conclusions

Ten WWS organizations were selected as good learning examples, and six of them were examined in depth in this study. They had made improvements and great progress in their work on climate adaptation and flood security "CA and FS". In the study, we clarified and concretized: (1) the influence of organizational form on success; (2) the driving factors, strategies, and other important explanations for success; (3) the challenges faced by the organizations; (4) the reasons why few organizations have an action plan.

WWS organizations in Sweden are not designed for the large investments required to cope with problems such as heavy rainfall [3]. Therefore, there is a need to disseminate the experiences of organizations that have focused on sustainability and have been successful with their efforts. One of the goals of this study was therefore to inspire other WWS organizations.

The documents of ten WWS organizations based on their SI results for the parameter "CA and FS" were studied. Then, seven organizations that had a value of 1.7 or higher on the weighting limits (Figure 3) were selected as good learning examples for this study, and six of them were in-depth interviewed.

Our results show that all six organizations studied have worked systematically to improve their work within the framework of "CA and FS". They differ in their topography,

size, and organizational form: three of the six are small and three are larger; two of them, Ängelholm and Luleå, are also coastal municipalities and the other four are inland; two of them, Arvika and Västerås, have a WWS company, and four, Värnamo, Jönköping, Ängelholm, and Luleå, have the traditional political board form. Three of them, Arvika, Värnamo and Jönköping, have been affected by torrential rains and floods in the past; the others know that disasters can occur at any time.

Regarding the influence of the organizational form, our results show that managers of both organizational forms, i.e., the traditional form that Ängelholm and Luleå have and the corporate form that Arvika and Västerås have, emphasize the positive role that their organizational form plays in their success.

They address almost the same factors that played a role in their success. These factors can be summarized as follows: they have short decision-making paths, can plan for the long term, and have managed to create interdisciplinary groups within their organization.

Previous research has found that both traditional and corporate forms have advantages and disadvantages for organizations [35,40] However, with the presence of active ownership in organizations, the conditions for long-term planning are created, regardless of the organizational form [37]. By active ownership, we mean here directors who give the management of the utilities a mandate to create the conditions for long-term sustainable governance [47]. These parties include both political and administrative leadership [48].

Thus, the six organizations studied clearly exhibit active ownership. This is evident in the support they receive from politicians and in the responses of the managers interviewed, as well as in the adaptation and flood control measures implemented. Therefore, both groups believe that their organizational form is the reason for their success.

Flooding events in Arvika, Värnamo, and Jönköping were cited as driving factors that advanced sustainability in climate adaptation issues. In the other three organizations, Ängelholm, Luleå, and Västerås, there were other driving factors and strategies that initiated and drove sustainability work. We believe that the following factors also have a significant influence on the promotion of sustainability in the six organizations studied: active ownership, knowledge resources to reduce vulnerability, and adaptive capacity to apply adaptation measures, in addition to technical and planning knowledge required in the application of robustness [28].

Some of the strategies used by the six organizations, as well as the main explanations for their success, were similar for some of them. In other organizations, however, there were unique explanations and strategies for each. For example, both Arvika and Jönköping took initial action for their low-lying areas. This was consistent with the recommendations of [42]: "The focus should be on actions in low-lying areas—the most vulnerable areas- to reduce the impact and risk of damage".

Ängelholm and Arvika formed a climate adaptation group and technical staff, respectively, with stormwater management professionals from different departments working together. The WWS organization in Västerås also cooperated with the City of Västerås, where there were other departments that shared the responsibility for stormwater management with them.

Examples of some unique strategies used by Luleå, Västerås, and Värnamo include the following: Luleå had adapted capacity building by collaborating with LTU and being part of an SWWA cluster (Dag och Nät). Jönköping used cloudburst mapping. Värnamo developed computer models to study how the measures would affect the town, and then created the action plan. In addition, Värnamo had applied for municipal grants from collective tax funds.

In this study, we also examined whether the organizations were managing the difficulties of sharing responsibilities and finances.

The study found that Arvika, Ängelholm, and Värnamo felt that they could handle task sharing well.

Ängelholm emphasized that the climate adaptation group found natural areas of contact between the different administrative entities. The two events that affected Arvika,

the extreme rainfall in 2000 and the floods in 2006, made all departments aware of the division of responsibilities and how to work together, and who should pay for each element.

In Jönköping, during the 100-years rainfall event in 2013, tax money was received, and the departments of roads and parks took their share of responsibility and took the necessary measures. However, Jönköping, Västerås and Luleå believed that sharing responsibilities was the most difficult challenge for them. This was in line with [49]: "research has shown that division of responsibilities is a complex issue and a barrier to strategic planning for urban water".

The study also found that both Jönköping and Västerås believed that clearer legislation would help solve the problem.

Regarding the reasons why few organizations (2%) had an action plan based on vulnerability analysis and were therefore green on the parameter (CA and FS), this study found that eight out of ten municipalities (Figure 3, Ta1) only had a vulnerability analysis and no action plan.

Thus, Jönköping and Västerås had an action plan and Värnamo are developing one. Arvika, Ängelholm, and Luleå did not have an action plan, since they did not consider it necessary to have one. The manager of Arvika believed that developing a detailed action plan would require frequent updates, distracting the organization from more urgent measures. Ängelholm and Arvika clarified that they had adequate control over the management of their network. Ängelholm and Luleå were not worried about the consequences of not having an action plan, since they were very close to the sea and had good recipients. Jönköping, on the other hand, has different geographical conditions because of Lake Vättern. Therefore, they had an action plan. However, in an earlier study [16], six managers gave other reasons for the lack of an action plan. Difficulty in sharing responsibilities was one reason.

To improve the performance of WWS organizations in "CA and FS", we suggest addressing, first and foremost, the problem of shared responsibility. This was the main challenge identified both in this study and in a previous study by [16]. According to the managers in Jönköping and Västerås, new legislations are needed. If so, the solutions lie with the government and the authorities. Nevertheless, some organizations (Arvika and Äengelholm) have succeeded in solving or mitigating the problem by, among other things, forming groups that include disciplines from several departments in the municipality, thus advancing the work with "CA and FS". The groups thus formed, and other attempts in Jönköping, Västerås, and Luleå, call for new legislation to help solve the problems, according to their managers.

We also believe that there are many other WWS organizations that have made improvements and progress on the parameter "CA and FS", as well as Arvika, Ängelholm, and Luleå, but they do not yet have an adopted action plan. Therefore, we believe that the SI assessment only 2% of WWS organizations are green on the parameter "CA and FS" at the national level, according to SI, is not the whole truth. Therefore, there should be other key factors in SI that measure performance in the parameter "CA and FS".

**Author Contributions:** N.N. designed and conducted the study, analyzed and interpreted the results, and wrote the article. K.M.P. contributed with several valuable ideas and options, supervised the entire study, and provided significant support. All authors have read and agreed to the published version of the manuscript.

**Funding:** The authors acknowledge the funding from the research program Mistra InfraMaint. Additional funding was received from the Department of Civil Engineering and Lighting Science, School of Engineering, Jönköping University.

**Institutional Review Board Statement:** Not applicable.

**Informed Consent Statement:** Not applicable.

**Data Availability Statement:** Not applicable.

**Acknowledgments:** The authors would like to express gratitude to Mistra InfraMaint for supporting the study by funding it. The authors offer special thanks to Magnus Bäckström, an expert at SWWA, for all the work he conducted to choose and contact the SWWA members. The authors wish to express special thanks to the interviewees, who devoted time and contributed valuable input and knowledge and contributed to what the study has achieved. We would also like to thank the Department of Construction Engineering and Lighting Science, School of Engineering, Jönköping University for funding this study and covering all practical costs needed for conducting the study. The project also received financial support from Mistra InfraMaint (grant decision 2018).

**Conflicts of Interest:** The authors declare no conflict of interest.

## Appendix A. SI Survey

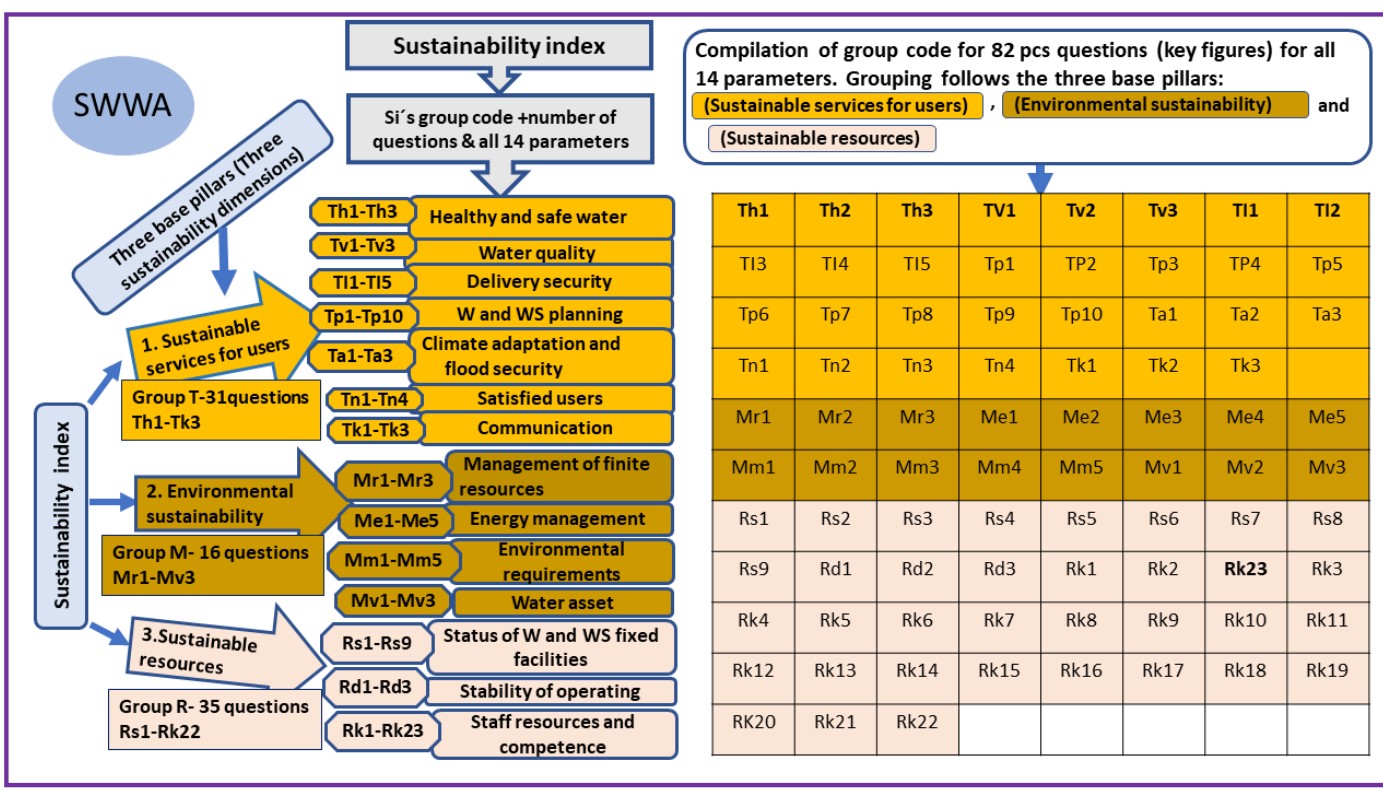

**Figure A1.** The 3 basic pillars, the 14 parameters, and the compilation of group codes for 82 questions [16].

## Appendix B. The Pictures of the Map in All Ten Municipalities

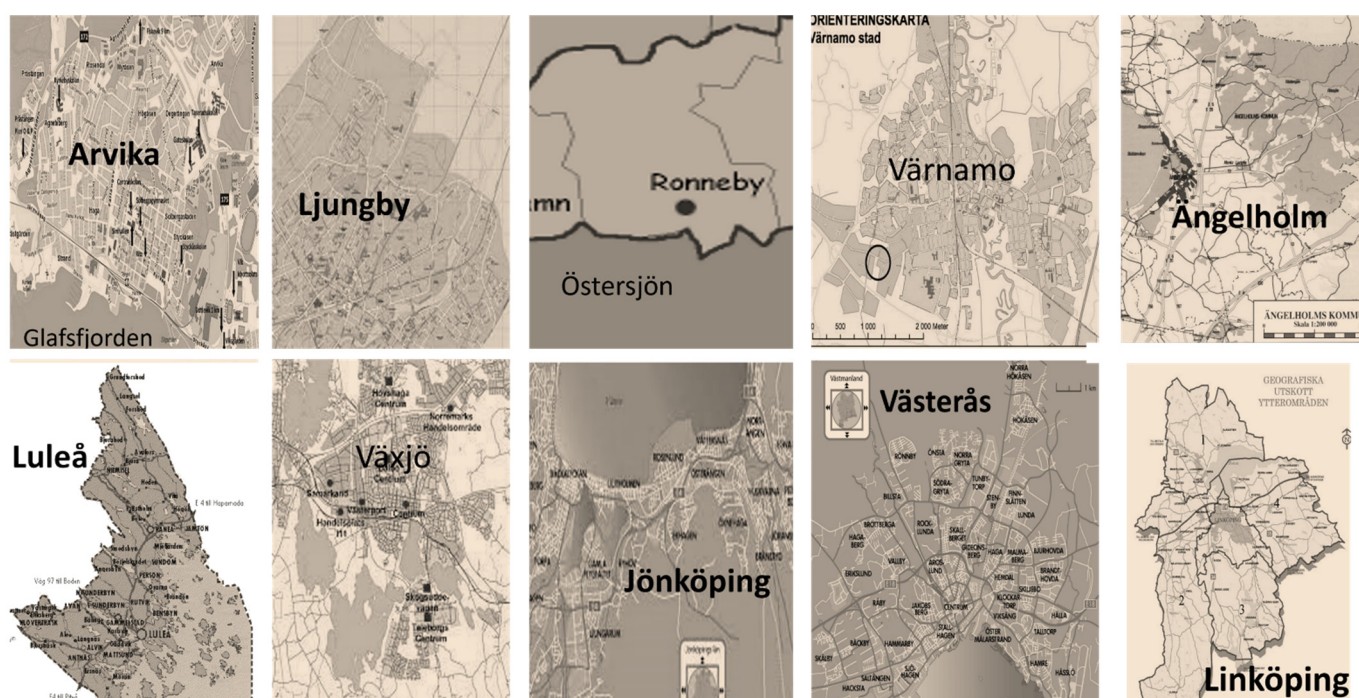

**Figure A2.** Pictures of the map in all ten municipalities.

## Appendix C

The Detailed Documents of the SI Evaluation of the Ten Studied WWS organizations.
The documents show all 14 parameters and 82 questions that make up the SI survey. Each parameter is shown as the main heading, and below the main heading are four columns. The headings of these columns are as follows: the code for the questions, the questions associated with each parameter, the communities' responses to each question and their color index, and comments, if any (Table A1). Figure A1 shows as an example of one of the 14 parameters in a typical detail in an SI evaluation document. It shows the code, the questions, and the answers with their color index score for the parameter "CA and FS" of the municipality of Arvika for 2020. It also shows the color index score for the parameter itself, based on the conditions of the evaluation (Table 1). Thus, in Table A1, the rating of the parameter is yellow because one of the answers is yellow.

**Table A1.** Part of a typical detailed document of the SI evaluation for the parameter "CA and FS" of the Arvika Municipality in 2020 [34].

| Parameter: Climate Adaptation and Flood Security (CA and FS). | | |
|---|---|---|
| **Code** | **The Questions** | **Answer with Color Index** |
| Ta1 | Is there an investigation with an action plan examining society's vulnerability due to more extreme rainfall and rising levels in seas, watercourses, and lakes? | Yes, but no action plan |
| Ta2 | Is there a clear strategy for new construction/reconstruction in terms of flood safety and correct height adjustment so that there is no damage to houses when the stormwater systems are overloaded? | Yes, and no floods can occur in new areas due to rain or water levels. |
| Ta3 | Basement floods within business areas as a 5-year average (the number per the coupling pipes of 1000 houses) | 0.21 |

## Appendix D

**Table A2.** The names of the organizations surveyed, the number of residents, the names of the managers of the WWS organizations, their number of years as managers, the number of years in the WWS sector, and the type of organization.

| Municipality/Number of Inhabitants | Manager/Investigator for WWS | Number of Years as the WWS Manager and in the Sector | Type of Organization |
|---|---|---|---|
| Arvika/25,865 | BA (Manager) | He has worked in the sector for 32 years and as a manager for 14 years. | "Teknik i Väst AB". A jointly owned company by Arvika and Eda municipalities in western Värmland. a merged organization" |
| Värnamo/34,530 | AV (WWS Manager) | He has worked in the sector for 15 years and as a manager for four years. | Traditional organizational form |
| Ängelholm/43,030 | RK (Network Manager) and (WWS Manager) | He has worked in the sector for 12 years and as a manager for five years. | Traditional organizational form |
| Luleå/78,487 | PV (WWS Manager) | She has worked in the sector for 14 years and as a manager for five years. | Traditional organizational form |
| Jönköping/142,630 | RR (WWS Manager) | He has worked in the sector for 17 years and as a manager for 13 years. | Traditional organizational form |
| Västerås/155,858 | AD (WWS Manager) JÖ (Investigator) | She has worked in the sector for more than 35 years and as a manager for five years. | "Mälarenergi AB" a Multi-utility municipally owned group company. |

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
