# Peer review of "Assessing Climate Adaptation and Flood Security Using a Benchmark System: Some Swedish Water Utilities as Good Learning Examples"

_water, doi:10.3390/w14182865_

Round 1

Reviewer 1 Report (New Reviewer)

The paper is intriguing and addresses a significant issue that is being exacerbated by climate change. The first major concern is the absence of a literature review, while the second is the presentation of certain paragraphs. There are numerous paragraphs that consist of fewer than two sentences. And I see no connection between the two paragraphs that follow. Therefore, I recommend that the authors revise the coherence and logic of the paragraphs and the content of this study. In addition, there are the following additional remarks:

1. The title of the paper Is not clear. I believe it should be revised to reflect the contribution of this study.

2. The keywords are not professional. More appropriate keywords should be defined for this study to reflect different aspects of this study.

3. Please revise lines 33 to 34.

4. Please use appropriate references to support line 35.

5. The logic of the introduction is not acceptable. Many paragraphs are aborted. I strongly suggest that the authors revise the introduction's logic, and in the introduction's final sections, the contribution of this study should clearly be highlighted.

6. This study suffers from the lack of a literature review. Relevant studies should be discussed. There are many studies that have explored different strategies such as mitigation, response, preparedness, ….. in the flood disaster management research area. For example, see: "An integrated decision model for managing hospital evacuation in response to an extreme flood event: A case study of the Hawkesbury‐Nepean River, NSW, Australia." Safety Science 155 (2022): 105867. And "A modelling framework to design an evacuation support system for healthcare infrastructures in response to major flood events." Progress in disaster science 13 (2022): 100218.

7. The authors have focused on Sweden, but the organisations which are discussed here are unfamiliar for readers. Therefore, the authors should provide a better perspective about the organisations named in this study.

8. The border around figures 1 and 2 ,.. are unnecessary

9. Figure 4 should be revised. Please add more labels to this figure

10. The research methodology should be presented in a more systematic way. In addition, while it is good to see that the authors have presented the research methodology as a flow chart, it is recommended that its presentation be improved.

Author Response

Comment 1:   The title of the paper Is not clear. I believe it should be revised to reflect the contribution of this study.

Author´s reply to comment 1. We are grateful to the reviewer for his comment. The new title, which better reflects the aim of the study, is as follows:

Assessing climate adaptation and flood resilience using benchmark systems: some Swedish water utilities as good learning examples.

Comment 2: The keywords are not professional. More appropriate keywords should be defined for this study to reflect different aspects of this study.

Author´s reply to comment 2. Thanks again to the reviewer for his comment. We have now defined new, more appropriate keywords. Line 29-31, (23-24)

Comment 3. Please revise lines 33 to 34.

Author´s reply to comment 3. The previous lines 33-34 are now revised as part of the revision of the entire "Introduction" Line 72-73 (65-66)

Comment 4. Please use appropriate references to support line 35.

Author´s reply to comment 4. Of course, a reference should be given. The reference is NO.(3) Svenskt Vatten (2016) "Stormwater, drainage water and wastewater: functional requirements, hydraulic planning and design of general drainage systems". Line 39 (32).

Comment 5. The logic of the introduction is not acceptable. Many paragraphs are aborted. I strongly suggest that the authors revise the introduction's logic, and in the introduction's final sections, the contribution of this study should clearly be highlighted.

Author´s reply to comment 5 We agree and many thanks for this comment. The introduction has been completely revised in accordance with the reviewer's suggestions, and the contribution of the study is now better emphasised. See lines 33–125 (26–117).

Comment 6. This study suffers from the lack of a literature review. Relevant studies should be discussed. There are many studies that have explored different strategies such as mitigation, response, preparedness, …. in the flood disaster management research area. 

Author´s reply to comment 6. We are grateful to the reviewer for this comment. Many relevant studies are now discussed and included in the introduction, in addition to the new sections 2. Literature Review. “Flood risk management: a shift from purely sectoral to integrated thinking." See lines 184-224 (118-158).

Comment 7. The authors have focused on Sweden, but the organizations which are discussed here are unfamiliar to readers. Therefore, the authors should provide a better perspective about the organizations named in this study.

Author´s reply to comment 7. Well, we agree with the reviewer that a better perspective should be given about the organizations. Therefore, we have tried to reorganize the "Case studies" section (Section 4) and added some more perspective. See lines 342-346 (267-271). Much more could also be interesting, but we think it would not enhance the manuscript.

Comment 8 The border around figures 1 and 2 are unnecessary

Author´s reply to comment 8. We agree with the reviewer and have now taken the border away.

Comment 9. Figure 4 should be revised. Please add more labels to this figure.

Author´s reply to comment 9. Figure 4 has been improved and more labels have been added to it. Line 3801 (298)

Comment 10. The research methodology should be presented in a more systematic way. In addition, while it is good to see that the authors have presented the research methodology as a flow chart, it is recommended that its presentation be improved.

Author´s reply to comment 10. Of course, and many thanks to the reviewer for this comment. The methodology is now improved by a few steps. Lines 235, 236, and 248 (169, 170, and 182)

Reviewer 2 Report (New Reviewer)

This authors detailed analyzed six WWS organizations and 52 SI annual documents to figure out the why they achieved success and what challenges they have faced. The lessons learned were used as reference for the other WWS organizations. Overall, this manuscript is well written, but a minor revision is needed before publication:

1.      What do you mean “One of of ten is green in parameter “CA” and “FS” in line 17?

2.      Figure 2 is not clear enough, please provide the one with higher revolution.

3.      What are the standards for parameter calculating and weight assignment in figure 3?

4.      More analysis of the parameter in figure 4 is suggested.

5.      Some recent relevant referecnes are missing, such as “Zhang, W.G., Phoon, K.K. Editorial for Advances and applications of deep learning and soft computing in geotechnical underground engineering. J. Rock Mech. Geotech. Eng. 4 (2022) 671-673” “Kok-Kwang Phoon & Wengang Zhang (2022): Future of machine learning in geotechnics, Georisk: Assessment and Management of Risk for Engineered Systems and Geohazards, DOI: 10.1080/17499518.2022.2087884”

6.      It is suggested to separate results and discussion properly to avoid getting a mess.

Author Response

The author's response to the comments in their entirety: we are grateful for all comments and remarks. They have really contributed to the improvement of our manuscript.

Note! The line numbers in brackets are valid after activating the function "Accept all changes and stop tracking".

Comments and author’s reviewer 2

Comment 1:   What do you mean “One of ten is green in the parameter “CA” and “FS” in line 17?

Author´s reply to comment 1. We are grateful to the reviewer for his comment. We have now explained the meaning better-see lines 21 (15-16). We have also explained it in lines 420-421 (336-337).

Comment 2:  Figure 2 is not clear enough, please provide the one with a higher revolution.

Author´s reply to comment 2. Of course, and many thanks to the reviewer for this comment. Figure 2 is now revised and has a higher resolution.

Comment 3. What are the standards for parameter calculating and weight assignment in figure 3?

Author´s reply to comment 3.  We agree with the reviewer that a better explanation should be given here, and it has been added to the report in lines 362-368, (283-289).  The explanation reads as follows: The statistical weighting method was used to evaluate the results, i.e., assigning a performance score for the parameter "CA and FS". Since in SI the answer to each question is given a traffic light colour, green (good), yellow (needs improvement), or red ( must be improved ), a performance score from 0 to 2 was used to calculate the weighting limits of the parameter. By setting a weighting limit of 2 for answers with green colour, 1 for yellow colour, and 0 for red colour, we calculated the weighting limits for the parameter as the average of the weighting limits of all questions belonging to that parameter

Comment 4. More analysis of the parameter in figure 4 is suggested.

Author´s reply to comment 4. Many thanks to the reviewer for his comment. In accordance with the reviewer's suggestions, Figure 4 has been improved and additional captions added. Line 380 (295). We have also mentioned in lines 379-380 (297-298) that the analysis of the results shown in Figures 3 and 4 is described in more detail in Section 5.1.  

Comment 5. Some recent relevant references are missing, such as “Zhang, W.G., Phoon, K.K. Editorial for Advances and applications of deep learning and soft computing in geotechnical underground engineering. J. Rock Mech. Geotech. Eng. 4 (2022) 671-673” “Kok-Kwang Phoon & Wengang Zhang (2022): Future of machine learning in geotechnics, Georisk: Assessment and Management of Risk for Engineered Systems and Geohazards, DOI: 10.1080/17499518.2022.2087884”

Author´s reply to comment 5. We are grateful to the reviewer for noting this. We have completely revised the introduction according to the reviewer's suggestions, including more than 20 additional references. We have also added a new "Literature Review" section (section 2).

Comment 6. It is suggested to separate results and discussion properly to avoid getting a mess.

Author´s reply to comment 6. This is a very important comment that also concerns a very important section of our manuscript, and we agree that it is often better to separate results and discussion. However, we have tried to interpret the results with a discussion of how to explain the results, so we would like to keep the discussion with the result as it is, for each research question and organization.

Round 2

Reviewer 1 Report (New Reviewer)

Congratulations! it can be published.

Author Response

Author's reply to the Academic Editor

We are sorry that we did not adequately answer all of Reviewer 2's questions. We thank you for your observations and patience. Below is the author's response, listed point by point. The revised texts are also highlighted in yellow in the attached file (water-1836327-minor revisions).
1- Sweden was added to line 11.
2- "Green" was explained in line 11 as follows: good performance level (green).
3- "Two" was written instead of "2" in lines 18 and 19.
4- Response to comment 6 by reviewer 2: see comment and response below.

Comment 6. It is suggested to separate results and discussion properly to avoid getting a mess.

Author´s reply to comment 6.

     The six long interviews with WWS managers yielded a large number of important responses from the interviewees, which answer the research questions very well and thus represent good results for the study. Therefore, the only way to get as much as possible out of the results is to discuss them in the same section. Furthermore, the nature of these results cannot be tabulated in a separate results section for later discussion in a discussion section. In that case, many of the results would have to be rewritten for discussion.
Therefore, in this study, it is very logical that we discuss the results to the extent that we report them for each research question so as not to add more length to an already large study because many results could then be repeated if discussed in a separate discussion section.
      However, we have better explained the structure of Section 5 with a few lines of text (lines 331-334). We have also made some additions to explain the results of Section 5.1 on Figures 3 and 4 in Section 4.1 (lines 291-295).

Regarding Ethical Questions Confirmation:
     Of course, you know very well the rules that apply, and we will of course follow them. Therefore, we have removed all the names of the WWS managers from the manuscript. We have also replaced the first initials of the managers' names in Table D1 in Appendix D with their names. However, we chose not to send the patient consent form to the managers for approval because we cannot control the time with them, and they may have other priorities. There may also be times when some of them are on vacation.

ATTENTION! The title of the article was changed at the request of reviewer 1 after a major revision. It is no longer "Good Swedish Learning Examples for progress in climate adaptation and flood security".

the new title is "Assessing Climate Adaptation and Flood Safety using Benchmark System: some Swedish Water Utilities as good learning examples".

This manuscript is a resubmission of an earlier submission. The following is a list of the peer review reports and author responses from that submission.

Round 1

Reviewer 1 Report

The good institutional management is indeed helpful for climate adaptation and flood security. However, this manuscript is too messy to understand the key findings. Most of the contents are related to the original reports and this makes it more difficult to read. I will suggest the authors to rewrite the manuscript and to summarize the results from the original reports, and then address the findings from analyses. The manuscript needs to be shortened and restructured. Descriptions about the ten selected cases are suggested to be added. The natural environment and precipitation might affect the decisions of the management policy. The background information might be the key factors contributed to the difference between organizations.

Reviewer 2 Report

The submitted paper concerns the important  issue of Swedish learning examples for progress in climate adaptation and flood security. The 184 municipal water and wastewater service organizations (WWS), 2% meet all benchmark requirements (i.e., obtain a green rating) on the parameter of climate adaptation and flood security set up by the Swedish Water and Wastewater Association's sustainability index (SI) in 2020. Ten WWS organizations were selected as good learning examples. The study highlights the strategies used and the key factors in the organizations' success. Remarks: Add English title of the references. The Figure D1. The risks associated with heavy rain in the combined system [2] in the appendix is obvious, you can omit it. Some guidelines about practical use of the obtained results should be presented. The last point of the article contains in fact only the conclusions relating to the researched case study, but there is no more detailed perspective. Figures and charts should have better quality, source files of them should be used, not like pasted figure 7 with some grids over it.  Instead of asterix * use x or “∙”. Authors should check the references, according to journal’s guidelines. Include the digital object identifier (DOI) for all references where available.

Reviewer 3 Report

This article is focused on showing good learning examples of municipal in dealing with flood security based on staff resources and competent in Sweden. It is a well written paper but due to narrow scope (local study) and collection of qualitative data (interviews), I think this study is better suited for a small commentary/mini review paper.